# Insecticidal characteristics and structural identification of the potential active compounds from *Streptomyces* sp. KR0006: Strain improvement through mutagenesis

**Young Sook Kim[1]◉, Mirjalol Umurzokov[2]◉, Kwang Min Cho[3], Jung Sup Choi[1]\*, Kee Woong Park[2,3]\***

**1** Eco-friendly and New Materials Research Center, Korea Research Institute of Chemical Technology, Daejeon, South Korea, **2** Department of Crop Science, College of Agriculture and Life Sciences, Chungnam National University, Daejeon, South Korea, **3** Daeseungbiofarm Co., Ltd., Daejeon, South Korea

◉ These authors contributed equally to this work.
\* jschoi@krict.re.kr (JSC); parkkw@cnu.ac.kr (KWP)

**Data Availability Statement:** All relevant data are within the paper and its Supporting information files.

## Abstract

Pest control by biological means is an effective, eco-friendly, and promising method that typically involves compounds naturally derived from actinomycetes. Thus, the present study aimed to screen, characterize, and identify the structure of insecticidal compounds from *Streptomyces* sp. KR0006 and increase the activity through mutagenesis. In the examination of the insecticidal activity level of the isolates, *Streptomyces* sp. KR0006 metabolite showed significant activity against larvae and moths of *Plutella xylostella*. Taxonomic analyses of the 16S rRNA gene sequences revealed that the isolated KR0006 strain tended to be 99% consistent with *Streptomyces cinereoruber* strain NBRC 12756. Three active compounds isolated from the culture filtrate of KR0006 were purified by solvent partition, midpressure liquid chromatography (MPLC), Sephadex LH20 column chromatography, and high-performance liquid chromatography (HPLC). By performing $^1$H-NMR, $^{13}$C-NMR, and 2D-NMR experiments, and high-resolution electrospray ionization mass spectrometry analysis, the 316-HP2, 316-HP3, and 316-HP5 compounds were inferred as antimycin A3a (MW, 519.; $C_{26}H_{36}N_2O_9$), antimycin A8a (MW, 534; $C_{27}H_{38}N_2O_9$), and antimycin A1a (MW, 548; $C_{28}H_{40}N_2O_9$) respectively. Mutant U67 obtained from exposure to ultraviolet (UV) irradiation (254 nm, height 17 cm) for 70 seconds resulted in a 70% more larval mortality than that of the initial wild culture. The second mutation of the culture broth enhanced insecticidal activity by 80 and 100% compared with the first mutation and initial medium, respectively. Our study found that *Streptomyces* sp. KR0006 strain produces insecticidal active compounds and could be used for practical pest management.

## Introduction

Chemical insecticides are most widely used to protect crop plants from harmful insects since they have high efficiency in pest control. At the same time, there is a common concern

**Funding:** This work was supported by Korea Institute of Planning and Evaluation for Technology in Food, Agriculture and Forestry (IPET) through Agricultural Machinery/Equipment Localization Technology Development Program, funded by Korea Institute of Planning and Evaluation for Technology in Food, Agriculture and Forestry (IPET) (321056-05). The funders had no role in study design, data collection and analysis, decision to publish, or preparation of the manuscript.

**Competing interests:** The authors have declared that no competing interests exist.

associated with the negative impact of insecticides on the environment, humans, and non-target organisms due to chemical pollution, accumulation of toxic residues in soil and water, and development of pesticide resistance in pests [1, 2]. In addition, the use of synthetic pesticides leads to the reproduction of insect pests and the proliferation of secondary pests and ultimately disrupts the ecosystem's ecological balance by killing natural pest predators [3, 4]. For these reasons, there is a need to develop and implement safe and environmentally friendly biological approaches to conventional chemical practices in pest management.

Bio-insecticidal metabolites as part of biological control in pest management have a great interest and are being broadly studied by researchers as they are eco-friendly, selective, biodegradable, and non-toxic natural products [3, 5–9]. Secondary metabolites of microbes are interpreted as low molecular weight substances [10], and this property is preferred in the development of biocontrol agents. Only in India, about 970 microbial compounds from at least 15 microorganism species were registered for use in pest management [11]. Entomopathogenic fungi such as *Beauveria bassiana*, *Metarhizium anisopliae*, *Lecanicillium lecanii*, *Nomuraea rileyi*, *Hirsutella thompsonii*, etc. showed excellent insecticidal activity [12–16]. For instance, *Isaria tenuipes* against the dengue vector *Aedes aegypti* (Linn.) and mycotoxins from *Beauveria bassiana* and *Metarhizium anisopliae* against *Spodoptera litura* larvae exhibited target-specific activity by blocking the activity of detoxifying enzymes such as carboxylesterase (α and ß) and superoxide dismutase [17, 18]. Among the biological control agents derived from microorganisms, actinomycetes also represent one of the most important and widely studied microbial resources because they produce various forms of active compounds for use as pesticides [19–21]. *Streptomyces* spp. is a major group of actinomycetes inhabiting the soil that shows insecticidal activity. An active compound derived from *Streptomyces diastatochromogenes* isolate showed significant insecticidal activity with $LC_{50}$ value of 10.26 mg/L against *Mythimna separate* [22]. Actinomycete isolates inhabiting the soil within the plants of the Asteraceae family in Pakistan showed 100% mortal activity against the first and fourth instar stages of *Culex quinquefasciatus* larvae [23]. Kim et al. [7] reported that active compounds isolated from *Streptomyces* exhibited insect growth regulatory and insecticidal activities against *Aedes albopictus* and *Plutella xylostella*. Gopalakrishnan et al. [24] also reported a novel metabolite purified from *Streptomyces* sp. CAI-155 has excellent larvicidal activity against second instar *Helicoverpa armigera* in both laboratory and greenhouse assays. Prasinons, avermectin doramectin, milbemycin, nanchangmycin, spinosad, and dianemycin have been derived as active compounds from the genus *Streptomyces* spp., against a variety of pest insects [25–28].

From the economic point of view, the production and application of the secondary metabolites from wild-type strains do not justify themselves much. Therefore, it is necessary to find ways to improve activity and achieve several times higher efficiency of the isolated secondary metabolites. A number of approaches such as mutagenesis, genetic recombination, selection, or improvement of fermentation processes have been studied and applied in practice to enhance the yield and activity of microbial strains [29–32]. Among them, the UV irradiation method is one of the most convenient, efficient, safe, and widely used in microbial strain improvement [30, 33, 34].

Thus, this study aimed to screen, characterize, and identify the structure of insecticidal compounds from *Streptomyces* spp. KR0006 and increase the activity through mutagenesis.

## Materials and methods

### Isolation of actinomycetes and fermentative incubation

A total of 850 *Actinomycetes* were isolated from the soil collected in the Jeongseon area, Gangwon-do, Korea using HV (humic acid-vitamin) and Bennet's agar medium. The Actinomycetes

strains were fermented in a 500 ml baffled Erlenmeyer flask containing 100 ml M3 culture media (6 g $CaCO_3$, 20 g glucose, 20 g soytone, 40 g soluble starch, and 1 L distilled water, pH of 6.9), which were incubated on a rotary shaker (250 rpm) at 27˚C for seven days. Fermentative broth cultures were subjected to centrifugation at 11,300 RCF (Relative Centrifugal Force) for 10 minutes and the supernatant was used for further study of insecticidal activity.

## Experimental insect

The diamondback moths (*Plutella xylostella)* and larvae were reared on cabbage leaves and maintained at 25˚C and 65% relative humidity with a 12 h light/12 h dark cycle.

## Bio-insecticidal assay

Insecticidal activities of actinobacterial culture media against *P. xylostella* larvae were evaluated using cabbage leaf disk dipping assay and whole plant assay. Chinese cabbage leaf disks (diameter 3 cm) were dipped for 30 seconds in an undiluted culture filtrate. Distilled water was used as an untreated control. Once the leaf disks dried, they were inoculated with *P. xylostella* 2nd instars (10 larvae/disk). The number of dead larvae was checked at 24 h intervals and calculated three days after treatment. In the case of the whole plant assay, 10 mL culture filtrate was sprayed on the foliage of Chinese cabbage (2 plants per pot) growing for 10 days under greenhouse conditions (30/25 ± 5˚C, light/ dark = 14/10 h). Then, plants were inoculated with *P. xylostella* 2nd instars (10 larvae/plant). The larvicidal and antifeeding activities were evaluated in the cabbage seedlings 5 days after inoculation (DAI). All experiments were independently repeated three times.

   The effect of ovipositional deterrent and controlling the larvae of *P. xylostella* was investigated using Chinese cabbage (40 seedlings per tray) grown for 10 days in a greenhouse. The culture filtrate 50mLwas first sprayed on the cabbage seedlings and the tray was put in a plastic box (30x25x30cm) and then inoculated with a diamondback moth (50 adults/box). At six DAI, the culture filtrate was secondarily applied to half of the tray. The fecundity and hatching rate and larval mortality were determined by comparing the first and secondary treatments with the untreated control. As a result of preliminary experiments, the activity of biological control agents (BCAs) *Bacillus thuringiensis* and chemical insecticides (Diflubenzuron) showed 100% insecticidal activity against diamondback moth when treated at the recommended treatment concentration (Table in S1 Table). All experiments were independently repeated three times.

## Taxonomic characteristics of the strain

To conduct taxonomic characteristics of the selected strain codded as KR0006, 16S rRNA gene nucleotide sequencing, and PCR amplification analyses were performed as reported by Bo et al. [35]. PCR products were commissioned to Macrogen Co., Ltd. (Daejeon, Korea) to perform gene sequencing. The sequences of the secured 16S rRNA gene obtained were analyzed through the BLASTn program of NCBI (http://www.ncbi.nlm.nih.gov) and were aligned with similar sequences using the CLUSTAL W program (MEGA 6.0). The phylogenetic tree of the strain was created using the Neighbor-joining method in the MEGA 6 program [36, 37]. The bootstrap analyses were implemented with 1000 replications to evaluate the stability of the constructed phylogenetic tree.

## Purification and structural identification of the active compounds

To separate the active substance produced by the strain KR0006, the culture filtrate(6L) was portioned twice with an equal volume of the Ethyl acetate (EtOAc) as reported by Kim et al.

[38] and concentrated *in vacuo*. EtOAc extract(1.85g) was purified by Mid-pressure liquid chromatography (MPLC) system eluted with a gradient of increasing methanol concentration (70% to 100%) in water, followed by Sephadex LH-20 (Phamarcia, Uppsala, Sweden) column chromatography eluted with 100% methanol.

The insecticidal activity of each obtained fraction was evaluated by conducting the leaf disk-feeding assay (see bio-insecticidal assay). The active fraction was finally purified by preparative HPLC on an ODS C18 reverse-phased silica gel column (Atlantis T3, 5$\mu$m, 10x250mm; WATERS) using 75% acetonitrile containing 0.02% trifluoroacetic acid over 60 minutes at a flow rate of 3ml minute$^{-1}$.

NMR spectra were recorded on a Bruker AVANCE HD 800 NMR spectrometer (800 MHz for $^1$H and 200 MHz for $^{13}$C) and Bruker AVANCE HD 700 NMR spectrometer (700 MHz for $^1$H and 175 MHz for $^{13}$C) at Korea Basic Science Institute (KBSI) in Ochang, Korea. NMR spectra were recorded in DMSO-$d_6$ and chemical shifts were referenced to the residual solvent signal ($\delta$C 39.5, $\delta$H 2.50). High-resolution electrospray ionization mass spectrometry (HR-ESIMS) data were acquired on a Q-TOF mass spectrometer (SYNAPT G2, Waters) at KBSI, Ochang, Korea.

### Strain activity improvement through Mutagenesis–UV mutation

The strain KR0006 spores was exposed to UV rays at a distance of 17 cm from the UV lamp (DESAGA, Sarstedt-Gruppe, MinUVIS UV Lamp) ($\lambda$ = 254nm) for 30, 45, 60, and 90 seconds. All these exposures were performed in a dark room to avoid any photoreaction in the production of mutants. The spores were then spread in plates containing Bennet's agar. The plates were incubated at 27˚C for 72 hours. The method employed for UV mutagenesis was described earlier by Khattab & Mohamed [39] and Wang et al. [40].

The survival and lethality rates were assessed after UV exposure and survived mutant colonies were cultured on an M3 liquid medium as described earlier (see fermentative incubation). Once mutant culture supernatants were harvested, the activity of each mutant was screened on the larvae of *P. xylostella* by conducting the leaf disk-feeding assay as described earlier (see bio-insecticidal assay).

The mutant strain that exhibited the highest insecticidal activity was selected and subjected to second-time mutagenesis aimed to increase the activity of the mutant obtained from the first mutation. The procedures of mutation and screening taken in the first mutation were carried out exactly the same way in the second mutation.

### Statistical analysis

One-Way ANOVA statistical analysis was performed using OriginPro 8.1 [41]. The Fisher's LSD test was used to compare the mean value, and the means values < 0.05 were considered statistically significant.

## Results

### Insecticidal activities of *Streptomyces* isolate

As a result of screening 850 actinomycetes for their insecticidal activity on the larvae of *P. xylostella* using the leaf disc feeding assay, high insecticidal activity (more than 80% mortality) was found in six actinomycete strains (KRA18-68, KRA18-104, KRA18-290, KR0512, KR0006 and KR0603) (Fig 1). Among these six strains, the KR0006 strain showed high insecticidal activities with mortalities of 100% after three days of treatment and was selected for further studies.

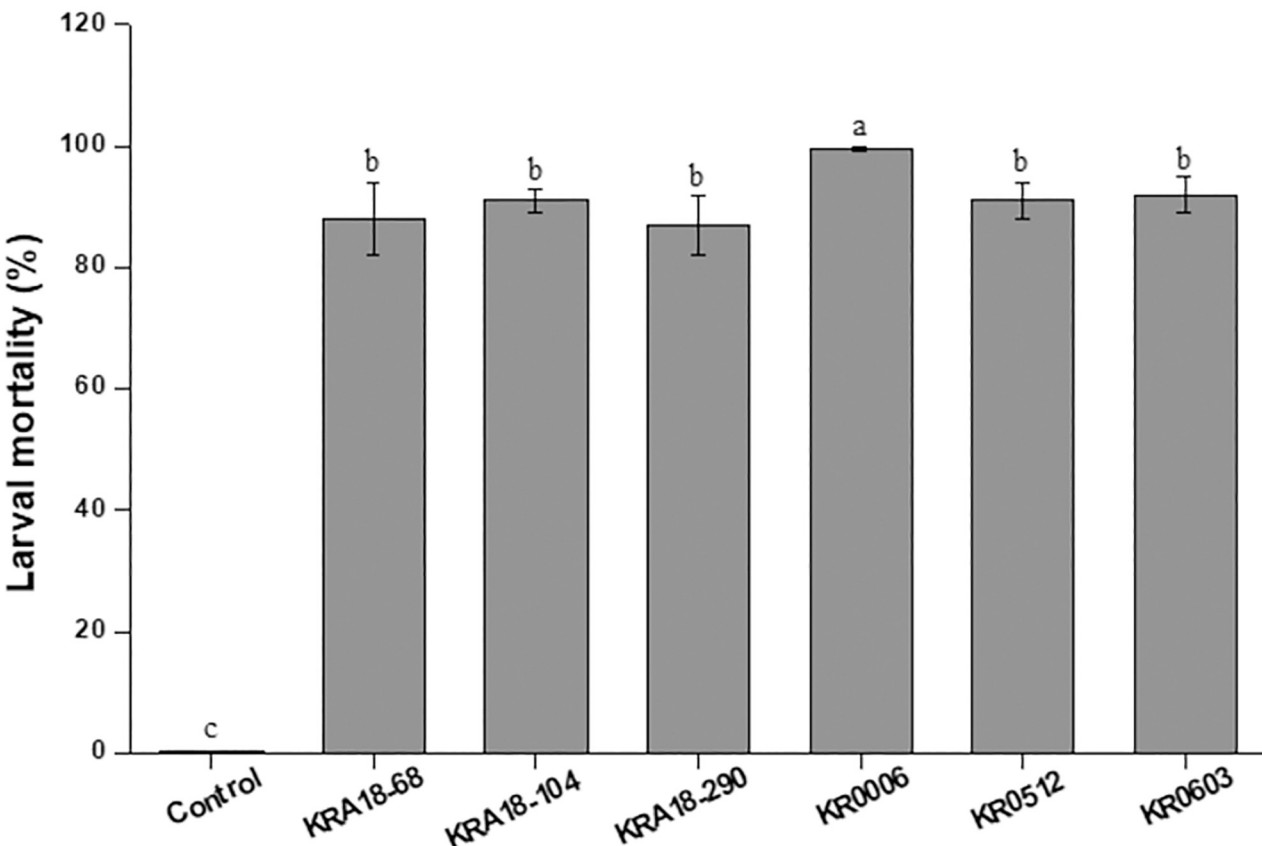

**Fig 1. Insecticidal activity of the actinobacteria strains against *Plutella xylostella*.** Second instar larvae of *P. xylostella* were treated with undiluted culture filtrate of each actinobacterium strain using the cabbage leaf dipping method. The mortality was evaluated three days after inoculation. The vertical bars indicate the standard errors of the means and different letters above the error bars indicate a significant difference in the Fisher's LSD test (P < 0.05).

In an experiment using whole plants, it was confirmed that the larvae control effect was 90% in 5-fold dilution at 5 DAI, but the mortality rate was significantly reduced when treated with 10-fold dilution. It was confirmed that the insecticidal activity of the KR0006 culture filtrate was concentration-dependent and the insecticidal effect increased with time after treatment (Fig 2). In addition, cabbage seedlings treated with even high-concentration of KR0006 culture filtrate, some feeding marks appeared on the cabbage leaves due to larvae activity at the initial stage of treatment, but the larvae were confirmed to be completely dead (Fig 3).

The oviposition deterrent activity evolution experiments revealed that the occurrence of the larvae was considerably less than that of the untreated control due to the inhibition of fecundity and the hatching of diamondback moths by the broth filtrate of KR0006. Also, the feeding marks caused by larvae after the secondary treatment of the culture filtrate were similar to those of the primary treatment, which is because the feeding ability of the larvae was inhibited by the secondary treatment of culture filtrate (Fig 4). Thereby, we confirmed that the culture filtrate of the KR0006 strain showed insecticidal activity by inhibiting the oviposition of moths and the growth and development of larvae.

### Taxonomic classification of the strain KR0006

The PCR technique using universal primers was performed to obtain the nucleotide sequence of the 16S rRNA gene of the KR0006 strain. An obtained nucleotide sequence was compared

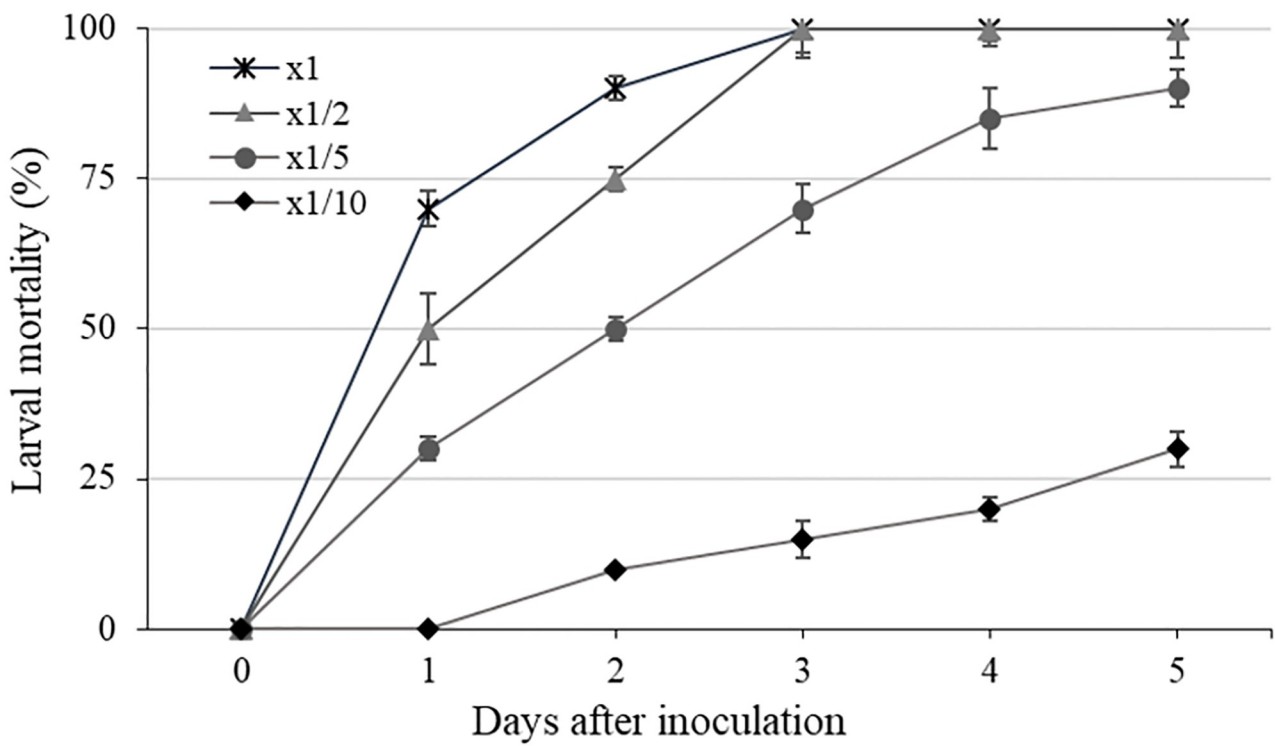

**Fig 2. Concentration-dependent insecticidal activities of KR0006 against *Plutella xylostella*.** The 10mL diluted solutions of the KR0006 were sprayed on the foliage of Chinese cabbage (2 plants) growing in the greenhouse. Plants were inoculated with *P. xylostella* 2nd instars (10 larvae/plant) and the mortality was calculated five days after inoculation. The vertical bars indicate the standard errors of the means.

with the nucleotide sequence of other *Streptomyces* strains through the BLASTn program of NCBI (http://www.ncbi.nlm.nih.gov). Results of comparative analysis with similar bacterial strains through 16S rRNA sequencing analysis showed that this strain tended to be 99% consistent with the *Streptomyces cinereoruber* NBRC 12756 strain (Fig 5).

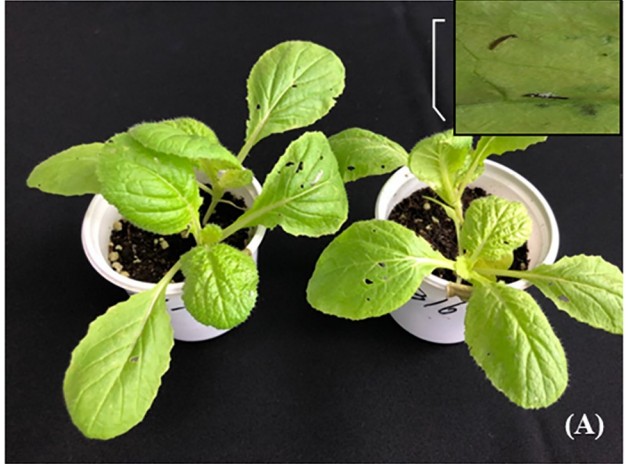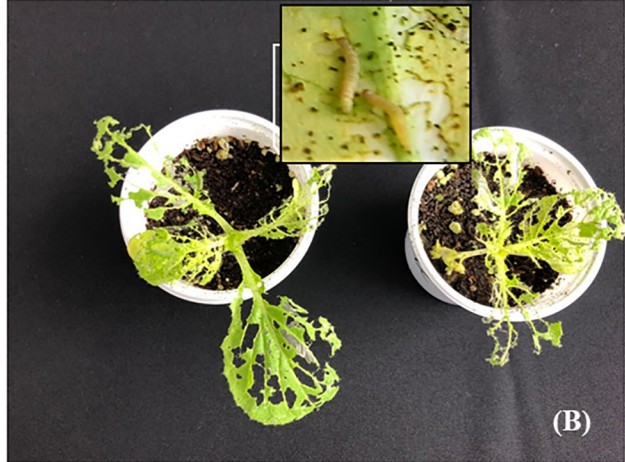

**Fig 3. Insecticidal activities of the broth filtrate of KR0006 against *Plutella xylostella* 2nd instars in greenhouse conditions.** Representative pictures were taken five days after inoculation. (A); 2-fold dilution of broth filtrate, (B); untreated control.

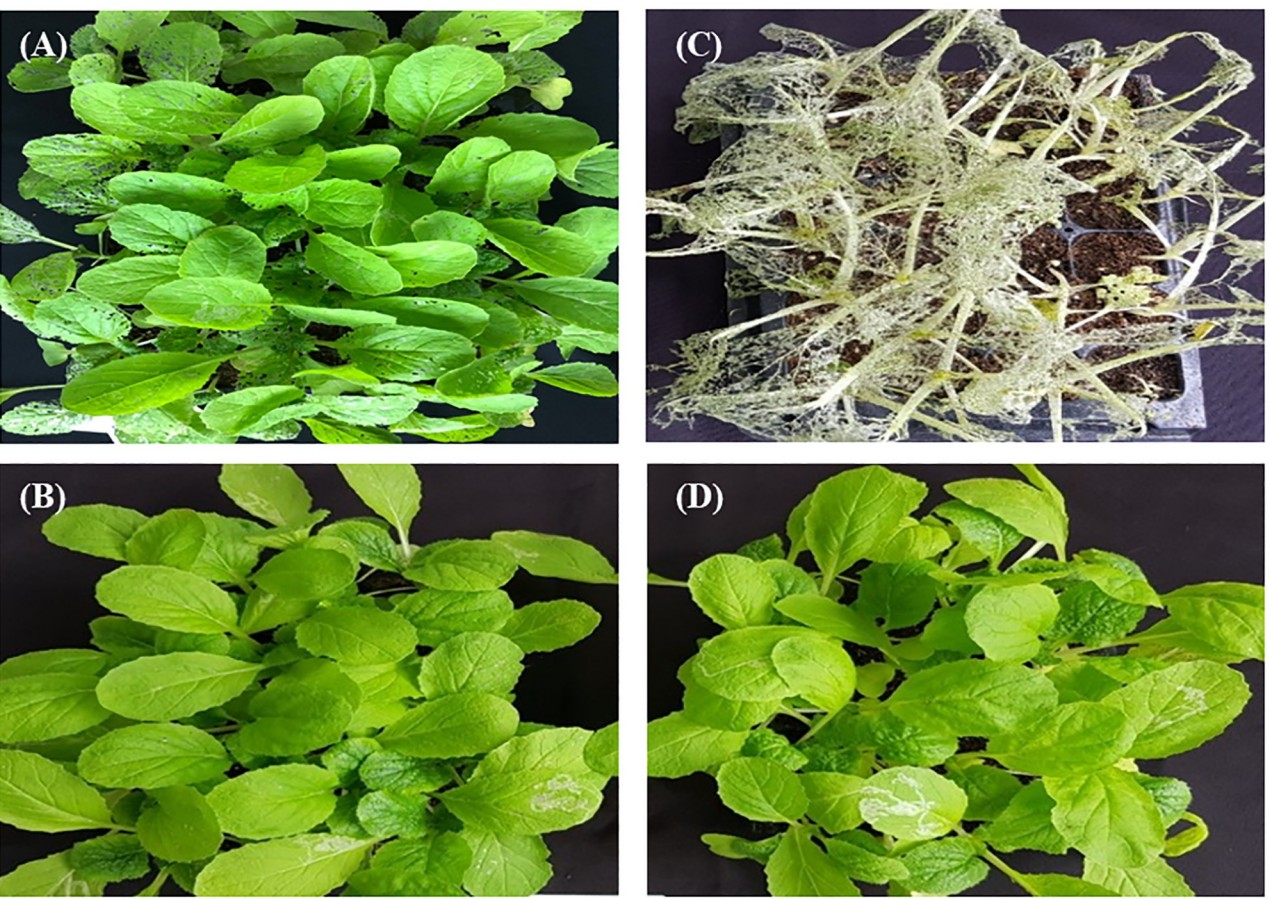

**Fig 4. The ovipositional repellency and larvicidal activity of *Streptomyces* sp. KR0006 against diamondback moth.** The culture filtrate was first sprayed on the cabbage seedlings grown in the tray, and then 50 diamondback moths were inoculated. Six days after inoculation (DAI), the culture filtrate was secondarily applied to half of the tray. (A); 6 DAI control, (B); 1st treatment of broth filtrate, (C); 10 DAI control, (D); 2nd treatment of broth filtrate.

## Purification and structural determination of the active compounds

The fermentation broth (10 liters) of strain KR0006 was partitioned between the EtOAc and $H_2O$. The EtOAc soluble portion was further separated by HPLC analysis using 75% acetonitrile containing 0.02% trifluoroacetic acid to yield five compounds 316-HP1–5 (Fig 6). Among the five compounds, the chemical structure of 316-HP2, 3, and 5 compounds that showed insecticidal activity against *P. xylostella* were determined by spectroscopic methods.

By performing $^1$H-NMR, $^{13}$C-NMR, and 2D-NMR experiments, and high-resolution electrospray ionization mass spectrometry analysis, the 316-HP2, 316-HP3, and 316-HP5 compounds were inferred as antimycin A3a, antimycin A8a, and antimycin A1a respectively (Fig 7). The structural features of antimycin A having methyl substituents at 2- and 6-positions, n-hexyl substituents at 8-positions, acyloxy substituents at 7-positions and an aroylamido substituent at 3-positions in the three active compounds are confirmed. Antimycin A is an optically active, nitrogenous phenol with the molecular formula $C_{28}H_{40}N_2O_9$ isolated from a culture broth of *Streptomyces* strain in 1949 [42]. The isolation of antimycins A1 to A16 have been reported so far, and each of antimycins A1~A8 is a mixture of two isomers containing a closely related alkylacyl group [43].

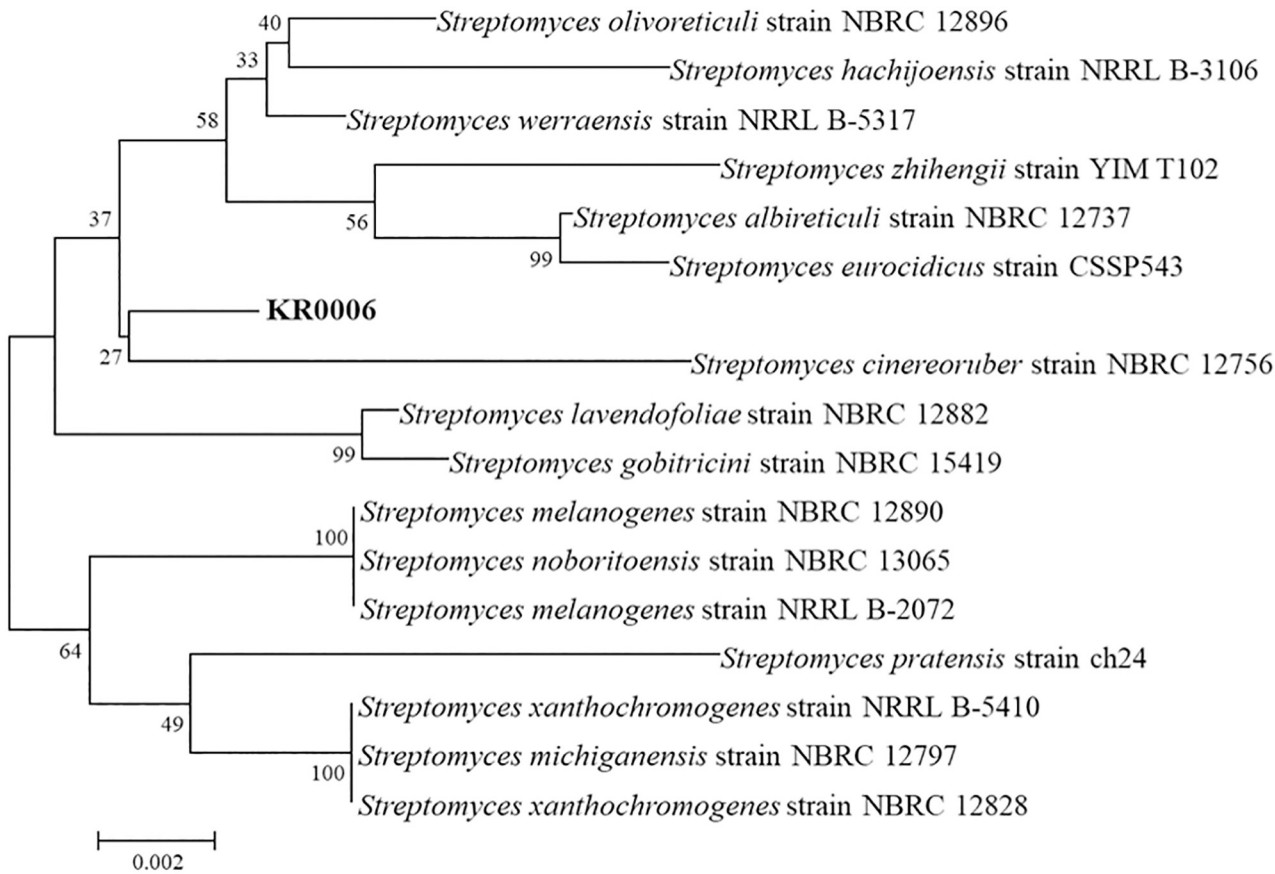

**Fig 5. Neighbor-joining tree of strain KR0006 based on its 16S rRNA gene sequence showing the phylogenetic relationship with related *Streptomyces* species.** Numbers at each branch node indicate the bootstrap percentage of 1000 replications.

For antimycin A8a (316-HP3), high-resolution mass spectrometry ([M+H]$^+$, *m/z* 535.2667; [M+Na]$^+$, *m/z* 557,2486) gave a molecular formula of $C_{27}H_{38}N_2O_9$. This requires that the alkyl side chain has a formula of $C_5H_{11}$. COSY connectivities from a methine at 2.53ppm (7-H) to methylene hydrogens at 1.65 and 1.36 ppm ($H_\alpha$), from Ha to methylene at 1.06ppm ($H_\beta$), from $H_\alpha$ to methane at 1.45 ppm ($H_\gamma$), and from $H_\gamma$ to a six hydrogen methyl doublet at 0.83 ppm ($H_\alpha$ and $H_\beta$), indicated that the alkyl side chain is isopentyl (3-methyl butyl). HMBC connectivities from $H_\delta$ and $H_\epsilon$ to $C_\gamma$ and $C_\beta$ support this side-chain structure [44]. The structure determination of antimycin A8 was first reported in 1997 by Barrow et al. [45] and was determined similarly to that of A7, and differed from antimycins A1 and A3 only at the 7-alkyl side chain.

Active compound 316-HP2 was inferred as antimycin A3a with the formula of $C_{26}H_{36}N_2O_9$ ([M+H]$^+$, *m/z* 520.2327). It was confirmed that this compound is significantly similar to the compound 316-HP3. However, there was no branched CH peak, as it appeared to be a linear alkyl in the side chain. The compound 316-HP5 with the formula of $C_{28}H_{40}N_2O_9$ ([M+H]$^+$, *m/z* 549.2819; [M+Na]$^+$, *m/z* 571.2637) was identified as antimycin A1a, a tautomer of the 316-HP3. Spectroscopic data for antimycins A1~A4 were in agreement with literature data [46–48].

### Strain activity improvement through mutagenesis–UV mutation

The number of mutant colonies decreased with the increases in UV exposure time. Among the tested UV mutants, the highest activity was witnessed in mutant U67. Larval mortality at five-

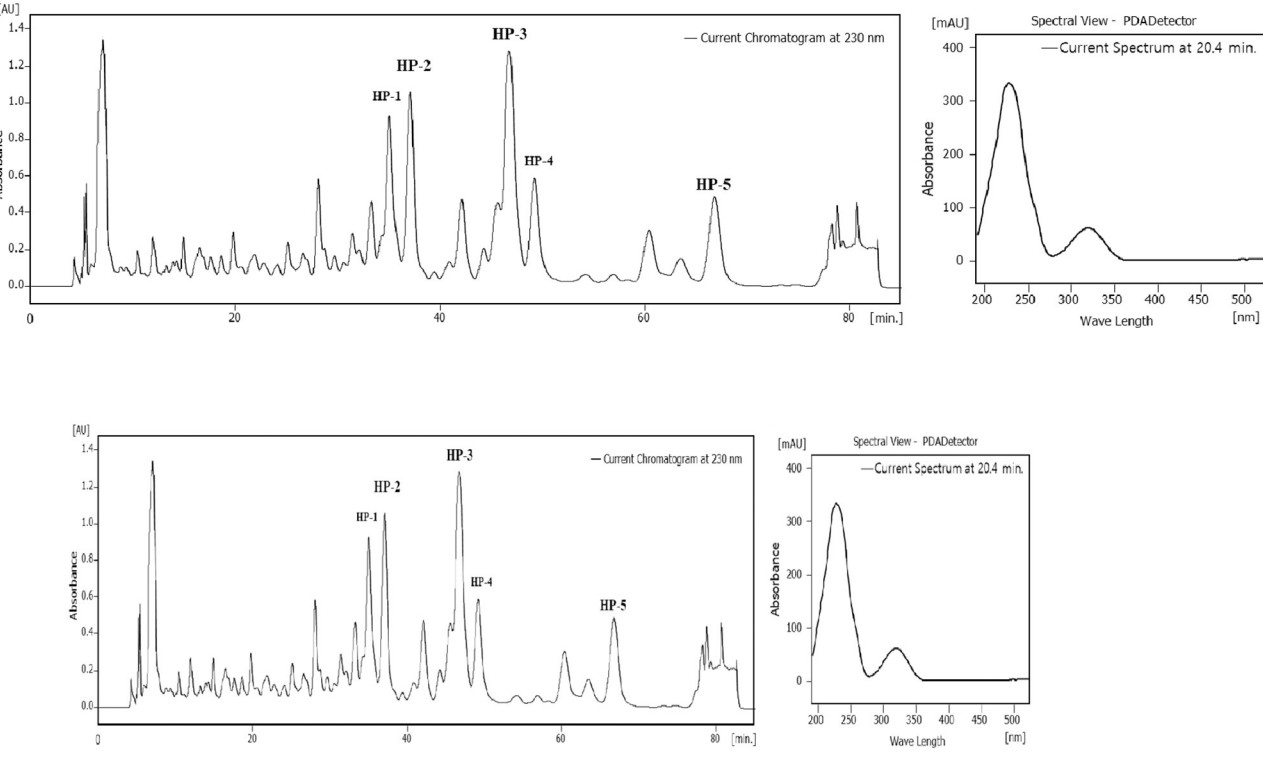

**Fig 6. High-performance liquid chromatography chromatogram of compounds 1~5 produced by *Streptomyces* sp. KR0006 and UV spectrum.**
Column, waters Atlantis T3 column (i.d. 4.6x250 mm); mobile phase, 70% acetonitrile/0.02% TFA; flow rate, 0.8 ml/min; detection, 230nm.

fold dilution broth filtrate of mutant U67 was 70% after 24 hours of treatment, and 100% after 48 hours of treatment, confirming that the insecticidal effect was higher compared to other mutants and wild type (Table 1).

In the secondary UV mutagenesis using mutant U67, the mutant U67-46 exhibited the highest larval mortality and was selected. The larvicidal activity of mutant U67-46 in the leaf disk feeding assay was 98% and 65% when treated with 20-fold and 30-fold dilutions, respectively (Table 2). In the case of mutant U67, the larval mortality was 55% when treated with 10-fold diluted culture filtrate indicating that the insecticidal activity of the mutant U67-46 was increased 3 times compared to the U67, and more than 5 times compared to the wild type (Table 2).

## Discussion

To overcome the unwanted effects of synthetic pesticides such as chemical pollution, accumulation of toxic materials, and development of pesticide resistance, studies for eco-friendly compounds as alternatives have been extensively conducted and continuously utilized [49]. More specifically, insecticidal, insect growth regulatory, and antifeedant activities of actinomycetes have been studied and confirmed that actinomycetes produce a range of chemical structures [4, 6, 50]. In this study, larvacidal activity against *P. xylostella* was investigated in 850 actinomycetes targeting the discovery of active compounds with insecticidal activity. The larval mortality was over 80% in the initially selected six strains, and among them, strain KR0006 showed the highest activity.

| | MW | R1 | R2 |
|---|---|---|---|
| Antimycin A3a (316-HP2) | 519 | $(CH_2)_3CH_3$ | $CH(CH_3)CH_2CH_3$ |
| Antimycin A8a (316-HP3) | 534 | $(CH_2)_2CH(CH_3)_2$ | $CH(CH_3)CH_2CH_3$ |
| Antimycin A1a (316-HP5) | 548 | $(CH_2)_5CH_3$ | $CH(CH_3)CH_2CH_3$ |

**Fig 7. The structures of the antimycins purified from *Streptomyces* sp. KR0006.**

Antimycins are known as secondary metabolites produced by *Streptomyces* and other types of soil bacteria and provide competitive advantages to kill neighbouring organisms such as insects, nematodes, fungi [51] and virus [52]. Several studies reported that the antimycin family of compounds has exhibited a broad spectrum of insecticidal and antifungal activities.

**Table 1. Insecticidal activity of the mutants against *Plutella xylostella* larvae.**

| Treatments | Larval mortality (%) | | |
|---|---|---|---|
| | After 24h | 48h | 72h |
| Mutant—U21 | 0±0e | 60±5c | 95±4a |
| U22 | 0±0e | 70±5b | 70±5b |
| U23 | 30±2c | 75±5b | 100±0a |
| U24 | 0±0e | 50±4c | 80±10b |
| U25 | 0±0e | 15±3d | 92±2a |
| U26 | 0±0e | 0±0e | 100±0a |
| . . .. | . . . | . . . | . . . |
| U67 | 70±5a | 100±0a | 100±0a |
| U73 | 50±5b | 80±10b | 100±0a |
| . . .. | . . . | . . . | . . .. |
| Wild type | 5±2d | 20±5d | 45±5c |

To estimate the larvicidal activity, Chinese cabbage leaf disks (diameter 3 cm) were dipped for 30 seconds in five-fold diluted (x1/5) each mutant and wild broth filtrate solutions. Once the leaf disks surface-dried they were inoculated with *P. xylostella* 2[nd] instars (10 larvae/disk). Values are the mean of three replication and different letters in each column indicate significant differences in the Fisher's LSD test ($P < 0.05$).

**Table 2. Insecticidal activity of KR0006 (WT) and mutants (first and second) against *Plutella xylostella*.**

| Dilution fold | Larval mortality (%) | | |
|---|---|---|---|
| | KR0006 (WT) | U67 | U67-46 |
| 1 | 100±0a | 100±0a | 100±0a |
| 2 | 100±0a | 100±0a | 100±0a |
| 5 | 90±3a | 100±0a | 100±0a |
| 10 | 15±4b | 55±3b | 100±0a |
| 20 | 0±0c | 25±3c | 98±2a |
| 30 | 0±0c | 0±0d | 65±10b |

Values are the mean of three replication and different letters in each column indicate significant differences in the Fisher's LSD test ($P < 0.05$).

Antimycin A9 isolated from fermentative broth of *Streptomyces* sp. K01-0031 exhibited insecticidal activity against *Artemia salina* and nematocidal activity against *Caenorhabtitis elegans* [53] and antimycins A10-16 isolated from fermentative broth of *Streptomyces* spp. SPA-10191 and SPA-8893 showed antifungal activity against *Candida utilis* [43]. Yan et al. [54] found the moderate fungicidal activity of antimycin A18 produced by *Streptomyces albidoflavus*. In our study, three active compounds identified as antimycin A3a, antimycin A8a, and antimycin A1a were purified from the culture broth of *Streptomyces* sp. KR0006. Our results coincided with recent results by Kim et al. [7] who had isolated five antimycins as active compounds from *Streptomyces celluloflavus* displayed significant insecticidal activity against *P. xylostella*, *A. albopictus*, *T. urticae*, and *F. occidentalis*. In addition to insecticidal and larvicidal activity, our results revealed that the culture broth of *Streptomyces* sp. KR0006 showed a considerable oviposition suppressing activity against the Diamondback moth. Although the antifeedant, larvicidal, and growth inhibitory activity of the secondary metabolites isolated from *Streptomyces* spp. has been reported [4, 7, 54, 55], the inhibition of oviposition has not been adequately studied yet. Taking into account high-level insecticidal, larvicidal, and spawn suppressing the activities of the strain KR0006, interest has arisen in conducting additional experiments of increasing the activity of this strain.

Strain improvement can be successfully achieved through conventional selection, UV mutation, or genetic recombination [30]. The activity or productivity of several microbial strains had been enhanced by UV treatment [56–59]. It became clear from our experiments that exposure to ultraviolet (UV) irradiation (254 nm, height 17 cm) for 70 seconds to KR0006 strain improved the larvicidal activity by 70%. There are very rare studies available on UV mutation for increasing the insecticidal or herbicidal activity of metabolites produced by *Streptomyces*. However, our results were closely in agreement with research finding where two times enhancement was observed in endoglucanase activity of UV treated *Aspergillus niger* strain [60] and up to 57.4% increase in *Streptomyces griseoaurantiacus* strain [58]. To achieve an even better activity, we subjected the selected mutant strain to the secondary UV irradiation which resulted in an improvement in insecticidal activity by 80 and 100% compared with the first mutant and initial wild strains respectively. There was no back mutation observed even after the second mutation which is common in most microbe mutation studies [56]. Our mutagenizes experiments confirmed that proper UV mutation can be forwarded as a tool to increase the activity of secondary metabolites produced by *Streptomyces*.

## Conclusion

In conclusion, *Streptomyces* sp. KR0006 containing antimycin A8a and its tautomers antimycin A3a, and antimycin A1a as active substances showed high insecticidal activity against *Plutella xylostella*. The culture filtrate of the KR0006 strain showed an insecticidal and larvicidal effect by inhibiting the oviposition of moths and the growth and development of larvae. The insecticidal activity of the culture filtrate was concentration-dependent. The yield of effective insecticidal substances produced by KR0006 was increased through mutagenesis by UV radiation. The insecticidal activity of 2nd mutant U67-46 increased 3 times compared to 1st mutant U67, and more than 5 times compared to the wild type.

In our study, *Streptomyces* sp. KR0006 was found to be a valuable biological control tool against diamondback moth (*P. xylostella*) and can be used to form an environmentally safe natural product in the ecofriendly pest management. However, further research is needed to make clear the mechanisms of insecticide action and evaluate the effectiveness in the open field.

## Supporting information

**S1 Table. Insecticidal activity of the biological control agent *(Bacillus thuringiensis)* and chemical insecticide (Diflubenzuron) against *Plutella xylostella* larvae.** To estimate the larvicidal activity, Chinese cabbage leaf disks (diameter 3 cm) were dipped for 30 seconds in the recommended concentration of insecticide solutions. Once the leaf disks were surface-dried they were inoculated with *P. xylostella* 2nd instars (10 larvae/disk). Distilled water was used as untreated control. Letters in each column indicate significant differences in the Fisher's LSD test ($P < 0.05$).
(PDF)

## Author Contributions

**Conceptualization:** Young Sook Kim, Mirjalol Umurzokov.

**Data curation:** Young Sook Kim, Mirjalol Umurzokov, Jung Sup Choi.

**Formal analysis:** Kwang Min Cho.

**Investigation:** Young Sook Kim, Mirjalol Umurzokov.

**Methodology:** Young Sook Kim, Jung Sup Choi.

**Resources:** Kwang Min Cho.

**Supervision:** Kee Woong Park.

**Visualization:** Jung Sup Choi.

**Writing – original draft:** Mirjalol Umurzokov.

**Writing – review & editing:** Young Sook Kim, Mirjalol Umurzokov.

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
