## [Decision Letter · Decision Letter 0]

14 Jun 2022

PONE-D-22-11701Insecticidal characteristics and s tructural identification of the potential active compounds from Streptomyces sp. KR0006: Strain improvement through mutagenesis.PLOS ONE

Dear Dr. Umurzokov,

Thank you for submitting your manuscript to PLOS ONE. After careful consideration, we feel that it has merit but does not fully meet PLOS ONE’s publication criteria as it currently stands. Therefore, we invite you to submit a revised version of the manuscript that addresses the points raised during the review process.

We look forward to receiving your revised manuscript.

Kind regards,

Sengottayan Senthil-Nathan, Ph D

Academic Editor

PLOS ONE

Journal Requirements:

"This work was supported by Korea Institute of Planning and Evaluation for Technology in Food, Agriculture and Forestry (IPET) through Agricultural Machinery/Equipment Localization Technology Development Program, funded by Korea Institute of Planning and Evaluation for Technology in Food, Agriculture and Forestry (IPET) (321056-05)."

Additional Editor Comments:

The manuscript need substation revision and reference section should be updated and discuss more about biopesticides including microbial pesticides in the introduction as well in the discussion at least the author should refer the following document s and rewrite the advantages of biocontrol potential of biopesticides. Please refer several articles published recently in Plos One.

I suggest the author to go through all the points suggested by reviewer and provide point by point explanation about your revision. In not please provide suitable rebuttal to avoid next round revision.

Introduction:

Please refer my first point, it needs more information about biopesticides through microbial culture with recent updates. # refer doi:10.3390/jof6040196, 10.1016/j.jip.2018.10.008, 10.1016/j.ecoenv.2019.109474, 0.1007/978-81-322-2056-5_3,

Result, discussion and Conclusion section needs to develop according to your result and appropriate citation must be provided to justify your claim

Reviewers' comments:

Reviewer's Responses to Questions

**Comments to the Author**

1. Is the manuscript technically sound, and do the data support the conclusions?

Reviewer #1: Yes

Reviewer #2: Partly

2. Has the statistical analysis been performed appropriately and rigorously? 

Reviewer #1: Yes

Reviewer #2: I Don't Know

3. Have the authors made all data underlying the findings in their manuscript fully available?

Reviewer #1: Yes

Reviewer #2: No

4. Is the manuscript presented in an intelligible fashion and written in standard English?

Reviewer #1: Yes

Reviewer #2: Yes

5. Review Comments to the Author

Reviewer #1: This manuscript presents “Insecticidal characteristics and structural identification of the potential active compounds from Streptomyces sp. KR0006: Strain improvement through mutagenesis.”. This manuscript formulated in clear and concise manner with well-supported statements. The problems arisen were well-addressed.

The regulatory mechanisms of these naturally derived compounds from actinomycetes were exhibited well enough, while these findings provide new insights how to control pests, which depends on biological approaches rather than chemicals.

The manuscript presented enough scientific novelties and my assessment is positive to publish it in the journal.

Reviewer #2: There are several aspects which need clarifications/implementation.

Please refer to the observations/suggestions available as notes in the enclosed file.

The authors should also check the following references since are related to the same topic:

1. Cao, Haijing, et al. "5′-Epi-SPA-6952A, a new insecticidal 24-membered macrolide produced by S treptomyces diastatochromogenes SSPRC-11339." Natural product research 33.5 (2019): 659-664.

2. Gopalakrishnan, Subramaniam, et al. "Insecticidal activity of a novel fatty acid amide derivative from Streptomyces species against Helicoverpa armigera." Natural Product Research 30.24 (2016): 2760-2769.

3. Tanvir, Rabia, Imran Sajid, and Shahida Hasnain. "Larvicidal potential of Asteraceae family endophytic actinomycetes against Culex quinquefasciatus mosquito larvae." Natural Product Research 28.22 (2014): 2048-2052.

6. PLOS authors have the option to publish the peer review history of their article (what does this mean?). If published, this will include your full peer review and any attached files.

Reviewer #1: **Yes: **Botir Khaitov

Reviewer #2: No

---

## [Author Response · Author response to Decision Letter 0]

6 Jul 2022

Journal Requirements:

• After carefully reading the formatting style, the authors followed wholly to meet the manuscript PLOS ONE’s style requirements.

2. Financial disclosure: 

• This work was supported by Korea Institute of Planning and Evaluation for Technology in Food, Agriculture and Forestry (IPET) through Agricultural Machinery/Equipment Localization Technology Development Program, funded by Korea Institute of Planning and Evaluation for Technology in Food, Agriculture and Forestry (IPET) (321056-05). However, the funder had no role in study design, data collection, and analysis, decision to publish, or preparation of the manuscript. This manuscript should be published in OPEN ACCESS. I will state the role of the Funder statement in the cover letter too and please change the online submission form on our behalf.

Additional Editor Comments:

3. The manuscript needs substation revision and the reference section should be updated and discuss more bio-pesticides including microbial pesticides in the introduction as well in the discussion least the author should refer to the following documents and rewrite the advantages of biocontrol potential of bio-pesticides. Please refer to several articles published recently in Plos One.

I suggest the author go through all the points suggested by the reviewer and provide a point-by-point explanation about your revision. In not please provide a suitable rebuttal to avoid the next round of revision.

Introduction:

Please refer my first point, it needs more information about biopesticides through microbial culture with recent updates. # refer doi:10.3390/jof6040196, 10.1016/j.jip.2018.10.008, 10.1016/j.ecoenv.2019.109474, 0.1007/978-81-322-2056-5_3,

The result, discussion, and Conclusion section need to develop according to your result and appropriate citations must be provided to justify your claim.

• Thank you for your valuable comments. The manuscript was revised and references and all the citations were also updated. In the introduction and discussion sections, the authors tried to discuss more detail bio-pesticides, especially microbial pesticides. Authors read all the literature suggested by the editor and reviewers and cited where needed. A number of articles published in PLOS ONE were also cited in the manuscript. Almost all the sections of the manuscript were rewritten and the manuscript was developed (refer to the manuscript file).

Review’s Comments to the Author

Reviewer #2: There are several aspects that need clarifications/implementation.

Please refer to the observations/suggestions available as notes in the enclosed file.

The authors should also check the following references since are related to the same topic:

Cao, Haijing, et al. "5′-Epi-SPA-6952A, a new insecticidal 24-membered macrolide produced by Streptomyces diastatochromogenes SSPRC-11339." Natural product research 33.5 (2019): 659-664.

Gopalakrishnan, Subramaniam, et al. "Insecticidal activity of a novel fatty acid amide derivative from Streptomyces species against Helicoverpa armigera." Natural Product Research 30.24 (2016): 2760-2769.

Tanvir, Rabia, Imran Sajid, and Shahida Hasnain. "Larvicidal potential of Asteraceae family endophytic actinomycetes against Culex quinquefasciatus mosquito larvae." Natural Product Research 28.22 (2014): 2048-2052.

• Thank you for your valuable comments. The authors read carefully the suggested papers by the reviewers and cited them where needed. The authors took into consideration the suggestions and comments in the enclosed file given by the reviewers and the manuscript was revised according to those.

---

## [Decision Letter · Decision Letter 1]

1 Aug 2022

PONE-D-22-11701R1Insecticidal characteristics and structural identification of the potential active compounds from Streptomyces sp. KR0006: Strain improvement through mutagenesisPLOS ONE

Dear Dr. Umurzokov,

Thank you for submitting your manuscript to PLOS ONE. After careful consideration, we feel that it has merit but does not fully meet PLOS ONE’s publication criteria as it currently stands. Therefore, we invite you to submit a revised version of the manuscript that addresses the points raised during the review process.

We look forward to receiving your revised manuscript.

Kind regards,

Sengottayan Senthil-Nathan, Ph D

Academic Editor

PLOS ONE

Journal Requirements:

Additional Editor Comments:

Minor revision is requested

Reviewers' comments:

Reviewer's Responses to Questions

**Comments to the Author**

1. If the authors have adequately addressed your comments raised in a previous round of review and you feel that this manuscript is now acceptable for publication, you may indicate that here to bypass the “Comments to the Author” section, enter your conflict of interest statement in the “Confidential to Editor” section, and submit your "Accept" recommendation.

Reviewer #2: (No Response)

2. Is the manuscript technically sound, and do the data support the conclusions?

Reviewer #2: (No Response)

3. Has the statistical analysis been performed appropriately and rigorously? 

Reviewer #2: (No Response)

4. Have the authors made all data underlying the findings in their manuscript fully available?

Reviewer #2: (No Response)

5. Is the manuscript presented in an intelligible fashion and written in standard English?

Reviewer #2: (No Response)

6. Review Comments to the Author

Reviewer #2: please refer to the observations available as notes in the enclosed file.

please refer to the observations available as notes in the enclosed file.

7. PLOS authors have the option to publish the peer review history of their article (what does this mean?). If published, this will include your full peer review and any attached files.

Reviewer #2: No

---

## [Author Response · Author response to Decision Letter 1]

10 Aug 2022

Journal Requirements:

• After carefully reading the formatting style, the authors followed wholly to meet the manuscript PLOS ONE’s style requirements.

2. Financial disclosure: 

• The funders had no role in study design, data collection, and analysis, decision to publish, or preparation of the manuscript.

3. The guidelines for resubmitting your figure files

• Thank you for showing us the guideline for adjustments of figure files. Now, all figure files are adjusted to the journal requirements and resubmitted.

Review’s Comments to the Author

1 80- it should be better to report the G value instead rpm

2 115- did you compare the results also with those available on Genebank?

3 205- you should compare the experimental data with those reported in literature for these compounds since are all already described. The relevant references should be provided.

4 209- if you are referring to the molecular formula it should be considered that the species [M+H]+ is the protonated form of and its empirical formula is different. It is formally one ion positively charged. It should be more clear.

5 Fig7- HMBC correlations cannot be checked without the experimental spectra (mono and bidimensional NMR)

• Thank you for your valuable comments. 

1 Rpm value of centrifuge was changed into g value in other word RCF (Relative Centrifugal Force)

2 Yes, obtained sequences of the secured 16S rRNA gene were analyzed through the BLASTn program of NCBI (http://www.ncbi.nlm.nih.gov).

3 Yes, we compared the experimental data with those reported in the literature for these compounds, and relevant references were cited where they are needed. Please refer to the manuscript file.

4 Thank you for your valuable comment. The structural determination part of the manuscript was deeply reviewed and revised. Please refer to the tracked manuscript or manuscript file to see corrections.

5 Figure seven is now provided with the experimental spectra (mono and bidimensional NMR).

---

## [Editor Report · Decision Letter 2]

4 Sep 2022

Insecticidal characteristics and structural identification of the potential active compounds from Streptomyces sp. KR0006: Strain improvement through mutagenesis

PONE-D-22-11701R2

Dear Dr. Umurzokov,

We’re pleased to inform you that your manuscript has been judged scientifically suitable for publication and will be formally accepted for publication once it meets all outstanding technical requirements.

Kind regards,

Sengottayan Senthil-Nathan, Ph D

Academic Editor

PLOS ONE

Additional Editor Comments (optional):

The manuscript can be accepted now
---

## [Editor Report · Acceptance letter]

8 Sep 2022

PONE-D-22-11701R2 

Insecticidal characteristics and structural identification of the potential active compounds from *Streptomyces* sp. KR0006: Strain improvement through mutagenesis 

Dear Dr. Choi:

I'm pleased to inform you that your manuscript has been deemed suitable for publication in PLOS ONE. Congratulations! Your manuscript is now with our production department. 

Kind regards, 

on behalf of

Prof. Sengottayan Senthil-Nathan 

Academic Editor

PLOS ONE